# Knee Health Is a Major Determinant of Mobility Across the Healthspan

**DOI:** 10.3390/jfmk10040454

**Published:** 2025-11-20

**Authors:** Brandon Marshall Peoples, Kenneth D. Harrison, Michael A. Samaan, Christopher B. Mobley, David T. Redden, Jaimie A. Roper

**Affiliations:** 1School of Kinesiology, Auburn University, Auburn, AL 36849, USA; bmp0049@auburn.edu (B.M.P.); kdh0077@auburn.edu (K.D.H.);; 2Department of Kinesiology & Health Promotion, University of Kentucky, Lexington, KY 40506, USA; 3Edward Via College of Osteopathic Medicine, Auburn University, Auburn, AL 36849, USA

**Keywords:** ACL, mobility, gait biomechanics, knee injury, osteoarthritis

## Abstract

Knee health constitutes a pivotal determinant of locomotor function and overall mobility throughout the healthspan. Impairments in knee function lead to a series of biomechanical and physiological alterations that pose substantial public health challenges and economic burdens worldwide. This review synthesizes evidence that delineates the complex and multifaceted relationship between knee health and mobility, emphasizing the interplay across various domains in adult populations. Knee health is conceptualized as a dynamic, interconnected system whereby structural integrity, neuromuscular mechanisms, biomechanical adaptations, and functional mobility are intricately interrelated through cascades of mechanistic processes operating across different temporal stages. A comprehensive understanding of these interrelationships is necessary for developing precise and targeted interventions that aim to preserve mobility and functional independence throughout the healthspan.

## 1. Introduction

Human mobility, the ability to move purposefully and efficiently through one’s environment [1], serves as a cornerstone of independence [2] and overall quality of life [3]. The ability to execute activities of daily living (ADLs), such as walking, stair climbing, and transitioning from a sitting to a standing position, facilitates societal participation, enables economic productivity, and supports well-being across the healthspan [4]. Healthspan refers to the period of life spent in good health, free from significant physical or cognitive limitations [4,5]. However, mobility limitations from musculoskeletal injury [6,7,8] or the natural aging process [3] initiate a cascade of negative consequences that can permeate every aspect of life, diminishing autonomy and increasing mortality. Consequently, the global rise in mobility limitations due to musculoskeletal injuries represents one of the most pressing public health challenges of our time [9].

Compromised knee health is a primary contributor to mobility limitations [6,7,10] and impaired locomotion [11,12,13]. Normal knee health relies on the congruency between the musculature, cartilage, ligaments, and bone [14,15,16,17,18,19,20,21,22,23] to meet the biomechanical demands of movement [24]. Compromised knee health due to acute injuries like anterior cruciate ligament (ACL) tears or chronic conditions like knee osteoarthritis (OA) disrupts the biomechanical and functional equilibrium of the knee, resulting in a cascade of detrimental effects. Substantial evidence suggests that muscle health declines [25,26,27,28,29], cartilage degrades [19,30,31,32,33], and bone remodeling occurs [18,19]. Furthermore, this alters knee biomechanics [17] and increases the mechanical and metabolic demands of locomotion [34]. Therefore, maintaining knee health means lessening mobility restrictions and preserving mobility.

The purpose of this review is to synthesize the current body of knowledge regarding the central role of knee health. This review will highlight that the knee is not a hinge joint, but a complex and sophisticated biomechanical system whose structural integrity, neuromuscular function, and sensorimotor systems are inextricably linked. Following an overview of the societal context of compromised knee health and the principles of healthy gait, this review examines the knee’s function as a pivotal junction in the lower limb kinetic chain. Next, we will explore the integrated systems of stability, including the sensorimotor system and the functions of the knee musculature. We will then transition to the pathophysiology of compromised knee health by examining the deleterious cycle between muscle health and cartilage degradation, the cascade of deterioration leading to compensations like the redistribution of joint kinetics, and finally, discuss how knee injury may impair knee power generation during locomotion. This review culminates in the proposal of an Integrated Knee Health Mobility Model (IKHMM) to guide future research in preserving mobility across the healthspan.

## 2. Methods

This narrative review was organized using a systematic search method to offer a thorough synthesis of the literature. We conducted searches on PubMed, MEDLINE, SPORTDiscus, and Google Scholar for peer-reviewed studies published from 1980 to 2025. Research Rabbit, a bibliometric assistance tool, was used to visualize connections between research articles and pinpoint additional key articles. The search combined keywords and MeSH terms such as “knee osteoarthritis,” “Knee OA,” “symptomatic knee OA,” “radiographic knee OA,” “anterior cruciate ligament,” “ACL,” “meniscus,” “knee injury,” “quadricep weakness,” “arthrogenic muscle inhibition,” “healthcare costs,” “sensorimotor system,” “proprioception,” “muscle function,” “biomechanics,” “gait,” “joint power,” “joint moment,” “total support moment,” “mobility,” “locomotion,” and “walking.”

The search mainly targeted titles and abstracts to identify relevant articles, followed by a full-text review to assess eligibility and synthesize findings. The inclusion criteria consisted of peer-reviewed articles in English that examined the connection between key aspects of knee health and mobility and the factors affecting adult human populations. Animal studies were only incorporated to provide mechanistic insights and were not the primary focus. Exclusion criteria included case reports, dissertations, conference abstracts, and non-peer-reviewed sources. To improve the reliability of the review, we emphasized “high-quality studies” and “robust methodologies,” such as systematic reviews, meta-analyses, large-scale longitudinal cohort studies (like the Osteoarthritis Initiative), and key mechanistic studies that establish foundational knowledge. Although this was not a formal systematic review, this approach aimed to minimize selection bias and develop a comprehensive, evidence-based foundation for the synthesis and the proposed theoretical model.

## 3. The Societal Context of Knee Health and Mobility

Mobility erosion poses a significant global public health challenge, with compromised knee health as a primary driver. A 2020 global population-based study estimated that approximately 654 million individuals aged ≥ 40 years suffer from knee OA [6]. Furthermore, the Global Burden of Disease initiative reported a startling 132% increase in years lived with disability due to knee OA since 1990 [7]. Years lived with disability measures how long individuals live with reduced health, functional limitations, and diminished quality of life. While the absolute burden is greatest in middle-aged and older adults due to the prevalence of degenerative conditions such as knee OA [6,7,35], a lifespan perspective reveals that the foundations of later-life disability are often laid decades earlier [36,37,38]. Kay et al. reported that, based on 20 studies (evidence level IV), 92% of 1156 pediatric and adolescent athletes (mean age 14 years) analyzed in their meta-analysis successfully resumed sports participation following ACL reconstruction [39]. However, the study also observed increased rates of re-injury and damage to the opposite knee within six years in this population, which elevates the risk of post-traumatic OA (PTOA) by adulthood [39,40]. Adults aged 18–39 years are disproportionately affected by sport-related injuries, which act as powerful catalysts for long-term joint decline through PTOA [36,41]. For instance, athletes in high-impact sports face a three-to seven-fold greater risk of developing poor knee health [42], and post-traumatic injuries, such as ACL and meniscus tears, establish a clear trajectory toward PTOA [36,37,38]. Longitudinal data reveal that women who suffer ACL injuries from soccer in their early 30s show a 51% rate of morphological knee changes within 12 years, indicative of declining knee health [37]. Conversely, 41% of men with horizontal meniscus tears in their late 30s developed signs of knee OA within 14 years [37]. This is further compounded by occupational hazards [43] and a significant burden on uniformed servicemembers who are predominantly young adults [41]. These distinct yet converging pathways underscore the fact that knee health is a concern throughout life. Thus, an injury in youth can initiate a decades-long process of subclinical deterioration that converges with age-related changes, creating a compounded burden later in their life.

### 3.1. Economic Impact of Compromised Knee Health and Comorbidities

The economic ramifications of compromised knee health are significant. Compromised knee health accounts for approximately 80% of the estimated $185 billion annual economic burden attributed to OA-related healthcare costs and lost productivity [44]. Direct costs, which include medical expenses for treatment and healthcare utilization, are substantial. Although not specific to poor knee health, there are significantly higher annual OA-related healthcare costs. Among insured adults, OA increased direct insurer expenses by $4833 for women and $4036 for men, with out-of-pocket costs rising by $1379 and $694, respectively [44]. Nationwide, this resulted in an estimated $185.5 billion in yearly OA-related spending, with women contributing a larger share due to higher prevalence and per capita costs. Indirect costs, which represent lost economic output from disability and reduced productivity, are equally impactful. Knee injuries represent a leading cause of lost workdays [43]. Isaacson et al.’s longitudinal analysis of military health records (2016–2021) revealed that knee injuries were the most common lower extremity musculoskeletal injury, resulting in over half a billion dollars in costs, a figure reflecting both treatment and lost duty time [41]. Additionally, the individual costs of meniscus repair or ACL reconstruction range from $12,000 to $72,000 [45,46], representing an increased financial burden that can reduce workforce productivity for younger adults, who are typically in the early stages of their careers. However, this economic burden extends beyond the direct costs of healthcare and indirect costs of lost productivity associated with poor knee health to include the societal costs of secondary healthcare expenses from reduced mobility. A 10-year longitudinal study found that middle-aged (45–64) and older adults (≥65 years) with lower extremity OA face a 53% increased risk of falls, escalating to 74% and 85%, respectively, when multiple knee or hip joints are affected [47]. This contributes to the approximate $20 billion in annual fall-related injury treatment costs among aging adults [48,49]. Adults with knee OA are also at a 51% risk of developing diabetes mellitus and a 53% risk of cardiovascular-related death compared with those without knee OA [8]. These higher risks for cardiometabolic diseases likely stem from lower physical activity participation, greater fatigability, and greater metabolic cost of movement [8,50]. Accordingly, understanding the factors that preserve knee health is critical for reducing individual and societal economic strain.

### 3.2. The Importance of Maintaining Knee Health

Age- and injury-related pathways leading to compromised knee health operate via distinct mechanisms; however, they ultimately converge to produce similar functional and economic consequences (see Figure 1). A comprehensive understanding of these divergent pathways will lead to the much-needed development of targeted preventive interventions tailored to various life stages and individual risk profiles. Ackerman et al.’s research highlights the need for more investigation in younger adult groups [36,51]. In the United States, increased population-based research utilizing healthcare datasets among younger adults is warranted to elucidate the factors contributing to the subsequent development of knee OA. Additionally, participation in sports has been identified as a significant factor influencing knee health [39,40]. According to Driban et al., there is a pressing need for comprehensive reporting of variables such as age, duration of participation in competitive sports, and history of joint injury to better ascertain which sports are associated with an elevated risk of knee compromise [42]. This is pressing, as the United States Department of Health and Human Services has issued a directive to increase the percentage of children and adolescents participating in sports to 63% from 50.7%, as many may continue to participate in sports well into adulthood [52]. Excessive mechanical stress or joint instability also contributes to this risk and plays a vital role in the workforce and military readiness. Isaacson and colleagues emphasize the need for preventative strategies tailored to the unit, setting, and mission of active servicemembers to reduce knee injury rates among military personnel [41]. Early departure from the workforce due to excessive mechanical stress or joint instability injury has implications that affect the gross domestic product through reduced workforce participation [53]. Moreover, loss of wages increases psychological distress, thereby impacting mental and overall health [51]. These findings demonstrate that maintaining knee health throughout life is vital for mitigating mobility disabilities, reducing economic burdens, and preserving military readiness.

## 4. Personal and Lifestyle Determinants of Knee Health

### 4.1. Age and Injury History

As previously mentioned, age is a significant factor in knee health, with multiple risks manifesting throughout the lifespan. Epidemiological evidence suggests that sports-related knee injuries sustained during young adulthood are associated with an increased likelihood of developing radiographically confirmed knee osteoarthritis (OA) in later life, with signs of OA detectable within one year of ACL reconstruction [54]. Although ACL reconstruction is associated with high rates of return to sport participation [39], it concurrently correlates with a 4.7-fold increase in the risk of developing moderate to severe knee OA [55]. In addition to sports-related knee injuries, any injury to the knee—including ligament, tendon, meniscus injuries, or fractures—results in approximately a sixfold increase in the risk of developing knee OA [56]. This risk is further amplified by the malalignment of the knee’s mechanical axis, commonly observed in conditions such as genu varum and valgus deformities [57]. While diagnosis for declining knee health may peak during late middle adulthood (age 55 to 65) [58], the path may be set decades prior by injury or malalignment. Taken together, these findings highlight the importance of maintaining knee health early in life to prevent long-term degeneration.

### 4.2. Obesity

Excess body mass is a significant and modifiable risk factor for knee health [59]. Elevated body mass index (BMI), particularly when coupled with lower limb malalignment, may increase mechanical loading on the tibiofemoral and patellofemoral joints, accelerating cartilage deterioration [60]. Besides mechanical influences, obesity is characterized by a state of chronic, low-grade systemic inflammation [59,61]. Adipose tissue in obese individuals secretes pro-inflammatory cytokines and adipokines that can contribute to cartilage degradation [59]. While several studies have shown that increased body mass is associated with increased risk of knee OA [59,62], the underlying mechanisms remain multifactorial and complex [63]. Abnormal mechanical loading, obesity-related malalignment, and muscle weakness may disrupt cartilage homeostasis and bone remodeling factors [60,64]. For a comprehensive review of obesity-related knee OA, see Chen et al.’s review on this topic [64].

### 4.3. Sex and Genetic Factors

Sex is a strong factor in declining knee health. Several studies have shown that females have a higher prevalence and more symptoms of poorer knee health than males, due to a mix of hormonal, anatomical, and biomechanical differences [65,66,67]. These include reduced cartilage volume, different joint loading patterns, and diverse inflammatory profiles [66]. Furthermore, distinct anatomical and biomechanical factors include a larger quadriceps (Q) angle (the angle formed by the pull from the anterior superior iliac spine to the center of the patella) and reduced muscle strength [66]. Larger static Q-angles are more frequently observed in women because of a wider pelvis and distinctive femoral and hip structures [68]. These anatomical features are associated with changes in patellar tracking and increase the risk of patellofemoral pain and dysfunction [57,66,69]. Besides the anatomical differences, hormonal changes during menstruation in female athletes and following menopause in older women can impact knee health [70,71]. For a comprehensive review of sex-specific factors associated with declining knee health, see Segal et al.’s review on this topic [66].

Genetic predisposition is a material contributor to knee OA risk. Recently, a large-scale Genome-Wide Association Study (GWAS) using UK Biobank and multi-omics data from nearly 140,000 participants with (~23,000 cases of radiographically confirmed and self-report hip and knee OA) identified nine new OA loci [72]. Zengini et al.’s findings demonstrated that hip and knee OA genetic expression profiles were highly correlated but distinct depending on joint-specific traits such as joint space width. For instance, they found that the single-nucleotide polymorphism (a variation at a single position in a DNA sequence) rs11780978 on chromosome 8 near the Plectin gene (PLEC) is associated with joint space narrowing [72,73]. Beyond these genetic variations, they established a causal relationship between high BMI, lumbar spine bone mineral density, and knee OA. Additional analyses revealed that common genetic variants explained about 15% of knee OA heritability, suggesting they may play a role in OA development. Please see Zengini et al.’s and Sorial et al.’s research for more details on the epigenetic variants that are associated with knee OA [72,73].

## 5. The Biomechanical and Energetic Foundations of Human Gait

Mobility depends fundamentally on effective locomotion, with walking as the primary mode of human movement in people living without any lower extremity disability. Human gait is a highly coordinated, repetitive pattern of limb movements that can be deconstructed into a fundamental unit known as the gait cycle [74,75]. This cycle, measured from the initial ground contact (heel strike) of one foot to the subsequent contact of the same foot, is divided into two distinct phases: stance and swing [74,75]. The stance phase, making up about 60%, is the period when the foot contacts the ground until it loses contact [75,76,77]. Stance is characterized by initial contact via heel strike to accept weight, stabilize and align the leg to support the contralateral leg toe off, and generate propulsive forces for forward progression [75,76,77]. During support, muscles play a significant role in counteracting the vertical ground reaction forces [76,77]. The remaining 40% is the swing phase, which involves advancing the limb forward with sufficient clearance to prepare for the heel strike [74,75]. This seemingly simple act requires the precise integration of neuromotor control [78,79], musculoskeletal coordination [76,77], and a constant stream of sensory feedback. Disruption of this cyclic locomotor pattern due to compromised knee health results in increased difficulty, elevated metabolic expenditure, and greater fatigability [50,80].

A defining characteristic of human bipedalism is its remarkable energy efficiency [81]. For instance, human walking is approximately 75% less metabolically costly than the bipedal and quadrupedal locomotion of chimpanzees [82]. This efficiency is achieved by leveraging passive dynamics, such as the pendulum-like motion of the swing leg [81], and the elastic energy storage and return of distal muscle-tendon units like the Achilles tendon [83]. Contrariwise, the stance phase is roughly three times as energy-intensive as the swing phase, highlighting the high metabolic demand of muscles for weight support and propulsion [76,77]. Donelan, Kram, and Kuo demonstrated that redirecting the center of mass during the stance phase necessitates significant mechanical work and metabolic expenditure, with longer stride lengths associated with increased energetic costs during step-to-step transitions [84]. Furthermore, their findings highlight the role of the individual limb contributions to locomotion. Although this observation was initially reported in a small sample (N = 9), numerous studies have since confirmed similar findings on individual limb contributions to mechanics and metabolic costs [85,86,87,88,89]. However, the energetic efficiency of bipedalism is not immutable. Compromised knee health leads to several subsequent neuromuscular adaptations [90,91,92]. Individuals with a history of ACL injury, for example, often exhibit persistent alterations in gait mechanics (including reduced knee flexion angles, diminished external knee moments, and slower walking cadences) that endure years after surgical reconstruction [90]. These deviations disrupt the coordinated limb dynamics that typically minimize energy expenditure, leading to increased metabolic cost during walking. Figure 1 from Landers-Ramos and Custer provides a clear visualization of these biomechanical changes, highlighting the altered joint kinetics and movement patterns that contribute to long-term joint degeneration [50]. Consequently, any pathological deviation undermines these efficient mechanics and forces greater reliance on active muscular control, which will inevitably increase the metabolic cost of walking [11].

## 6. The Knee as a Biomechanical Nexus in the Kinetic Chain

Effective locomotion depends on the proper function of the knee, which bears weight and acts as a junction connecting the hip and ankle. To understand this role, one must first consider the principle of the kinetic chain, as first described by Franz Reuleaux, which represents the body as interconnected segments where force applied to one segment affects the others [93]. During walking, ground reaction forces are transmitted from the foot proximally up the chain through the ankle, knee, and hip. In contrast, forces generated by the powerful hip and trunk musculature are transmitted down the limb to control mediolateral movement of the center of mass and propulsion [24,75,94]. Positioned anatomically and functionally between the hip and ankle, the knee acts as a pivotal fulcrum for this bidirectional management of ground reaction forces [24]. The knee’s function is not that of a simple hinge, but of a sophisticated biomechanical mediator [24]. Through controlled flexion and extension, the knee modulates limb length to absorb shock during weight acceptance and ensures adequate foot clearance during swing [24,74,81]. By varying its stiffness via muscle co-contraction, the knee actively dissipates impact forces, protecting the hip and spine, while also fine-tuning the delivery of propulsive power from the hip down to the ground [24]. Thus, a stiff, painful, or unstable knee loses this buffering capacity, leading to inefficient compensatory movement and the transmission of potentially damaging transient impulsive forces throughout the kinetic chain. Muscle co-contraction and its role in compromised knee health will be further discussed in the section on muscle weakness as a driver of cartilage degradation.

Importantly, not all challenges to knee function arise from within the joint itself. Structural asymmetries stemming from the femur or tibia can impose external biomechanical demands that hinder the knee’s ability to absorb ground reaction forces during gait, contributing to persistent clinical symptoms [95,96]. Leg length inequality or discrepancies greater than 20 mm (2 cm), whether measured anthropometrically with a tape measure or radiographically using digital rulers, is associated with increased pain, stiffness, aching, and joint degeneration at the knee [95,97,98,99]. Notably, current evidence suggests that the knee may be the initial site of degradation within the kinetic chain, precipitating compensatory degradation in the hip or lower back [96,99,100]. Leg length asymmetry affects both adolescents returning to sport after ACLR [98] and older adults with knee OA [97], underscoring its significance as a critical factor in the knee’s kinetic role.

### 6.1. The Stability of the Knee: A Synthesis of Passive and Active Systems

The stability of the knee during locomotion is provided by multiple structures through an integrated system of passive and dynamic viscoelastic (combination of viscous fluid-like and solid elastic-like properties) components working in harmony [101], as illustrated in Figure 2. The fundamental structural integrity of the knee is provided primarily by four passive viscoelastic structures: ACL, Posterior Cruciate Ligament (PCL), Medial Collateral Ligament (MCL), and Lateral Collateral Ligament (LCL) [102]. These viscoelastic passive structures act as vital restraints to excessive joint motion [101,103,104,105]. However, these passive structures are insufficient to stabilize the knee against the high dynamic loads of walking or running [77]. This requires dynamic stability, an active process whereby muscles crossing the joint generate compressive forces and produce torques (moments) that resist motions that would otherwise strain the ligaments. For example, a well-timed hamstring contraction can pull the tibia posteriorly, reducing strain on the ACL [106]. Thus, the ligaments define the safe envelope of motion, and the muscles provide the real-time enforcement of these boundaries [101].

### 6.2. The Sensorimotor System of the Knee

The sensorimotor system, a complex neurological system that allows the body to sense joint position and proprioception [107,108], links the passive viscoelastic structures (ligaments and tendons) and dynamic viscoelastic structures (muscles) to react and maintain joint stability [101,103,104,105]. This joint sense is derived from a constant stream of information from specialized mechanoreceptors (e.g., Ruffini endings, Pacinian corpuscles, Golgi tendon organs) located within the joint capsule, muscles, tendons, and, critically, the ligaments themselves, as illustrated in Figure 3 [101,107,108]. When a ligament is stretched, its embedded mechanoreceptors send afferent feedback to the central nervous system (CNS) [101,105]. Furthermore, the CNS rapidly processes this signal and generates an efferent motor command, triggering a reflexive muscle contraction that counteracts the destabilizing motion and protects the ligament [101]. These neurosensory regulatory processes transform ligaments from simple passive tethers into active sensory organs, which have profound implications for injury, especially in fatiguing situations [107,109]. For example, an ACL tear or knee OA is not just a mechanical failure but also a significant neurosensory injury that alters a critical population of mechanoreceptors, disrupting the sensorimotor loop [110,111,112]. Accordingly, this loss of proprioceptive information leads to persistent deficits in neuromuscular control, which helps explain non-contact injuries and why people with ACL injuries often suffer from functional impairments and a high rate of reinjury.

### 6.3. The Knee’s Role in Managing Ground Reaction Forces

The knee’s ability to manage the complex mechanical demands of locomotion depends on the precise and powerful actions of its surrounding musculature [74,94]. When the heel strikes the ground during stance, it generates ground reaction forces that create external moments (rotational forces) that would cause the ankle, knee, and hip to collapse if not opposed. The musculature surrounding each joint generates internal moments to counterbalance these external moments, thereby maintaining stability and controlling movement [76]. For the knee, the knee extensors (quadriceps) are the primary muscles responsible for regulating the knee during critical eccentric and concentric contractions during locomotion [24,76,77,79]. At heel strike, the vasti group (vastus intermedius, vastus lateralis, and vastus medialis) eccentrically contracts to control knee flexion, acting as a brake to absorb the percussive forces from striking the ground [76,77,78]. Evidence from musculoskeletal modeling and muscle synergies research agrees with this muscle activity during normal and fast walking speeds [76,77,78,79]. This function is paramount for protecting joint cartilage, as it reduces shear forces on the ligaments that stabilize the knee capsule [101]. Through midstance, the monoarticular vasti group contracts concentrically and then isometrically to extend the knee and support the body as the contralateral limb goes through its swing phase [77]. Alternatively, the knee flexors (hamstrings), a biarticular muscle group, play a dual role in locomotion [77]. In late swing, they contract eccentrically to reduce the angular velocity of the forward-swinging leg, preparing it for landing [77,113]. Additionally, during the early stance phase, these muscles assist the glute muscle group in hip extension to propel the body forward [76,77]. While the demands of walking are primarily in the sagittal plane (flexion-extension), this does not fully capture the knee’s role in dynamic stability. Rotational (transverse plane) and side-to-side (frontal plane) motions must be controlled to prevent injury, especially during cutting or pivoting [114]. This dynamic stability is provided by a concert of muscular action by several accessory muscles [115]. This multi-planar neuromuscular control is important for maintaining joint congruency and preventing excessive strain on passive structures, such as the ACL [101].

Within the knee joint system, the vasti muscles play an important role in modulating lower limb support and progression during weight acceptance and stance, with their activity changing dramatically in response to walking speed. At slower speeds (categorized as very slow and slow), the body employs a gait pattern characterized by a more extended, or straighter, leading limb during early stance [77,113]. This posture allows skeletal alignment to provide much of the vertical support against gravity (Figure 4A), functioning much like a passive strut and minimizing the need for active muscular contribution from the vasti [77,113]. For instance, the knee flexion angle during early stance is reduced to approximately 10° during slow walking compared to around 23° at a freely chosen walking speed, as illustrated in Figure 4C. As speed increases, from slow to free, a significant shift in gait mechanics occurs. Additionally, this transition is marked by greater stance phase knee flexion, which fundamentally alters the demands of the knee extensors [77,78]. The increased compliance of the flexed knee requires a dramatic increase in force from the vasti muscles to provide vertical support and prevent the limb from collapsing. However, this necessary supportive function comes with a biomechanical trade-off. The same increased vasti force that provides support also acts to resist or slow forward progression during early stance (see Figure 4B). This braking effect highlights the dual role of the vasti and higher walking speeds: they are necessary to mitigate impact forces and stabilize the knee, but in doing so, they also generate a force that opposes progression. Because of this speed dependence, this underscores that the vasti are not simply extensors, but are critical modulators of gait, with their contribution to both support and progression being highly sensitive to the overall kinematic strategy adopted at different walking speeds.

### 6.4. Changes in Joint Kinetics in Adults with Compromised Knee Health

The functional decline associated with compromised knee health is intimately linked to the power-generating capacity of the knee extensor muscles [77]. Knee injury and the subsequent disuse and inflammation are known to trigger significant changes in muscle fiber composition, including a shift from powerful, fast-twitch type IIa fibers to less powerful hybrid type IIa/x fibers, leading to muscle weakness [26,29]. Muscle weakness will be discussed in detail later. Consequently, the ability to shift from concentric to eccentric power during initial contact and weight acceptance is reduced in individuals with compromised knee health, depending on their functional status [116]. Joint power reflects the rate of energy transfer at a joint and is calculated as the product of the moment and the angular velocity [74,117]. Positive power indicates energy generation through concentric muscle actions, while negative power reflects energy absorption via eccentric control [117]. Winter first determined that knee power during gait can be divided into four distinct phases: K1: a period of increased eccentric power absorption from knee extensor activity during the loading response, K2: a period of concentric power generation during midstance, K3: a period of eccentric activity during the pre-swing phase, and K4: a period of eccentric activity during late swing (see Figure 5A) [75]. Impairment in the K1 phase results in a diminished K2 concentric power output, contributing to a less powerful (see Figure 5B), less stable, and potentially more metabolically costly gait, although this has not been extensively studied. However, recent findings from a cross-sectional analysis of joint power (N = 629) conducted by Liew et al.’s indicate that K1 power does not significantly vary with gait speed, suggesting its potential as an age-related biomarker. More research is warranted to determine its role in compromised knee health.

These alterations in joint powers are accompanied by a redistribution of joint moments across the lower limb kinetic chain, which reflects a hallmark compensatory strategy of the neuromuscular system [118,119,120,121,122,123,124,125]. This redistribution is evident in older adults with knee OA. For instance, Zeni and Higginson, in a cross-sectional study of 30 individuals with knee OA and 15 healthy controls, found that while the total support moment (the sum of moments from the ankle, knee, and hip during stance) [94] remains similar to healthy controls, the contribution from the arthritic knee to the total support moment is significantly reduced [126]. To compensate, individuals with knee OA usually increase the ankle and/or hip. Zeni and Higginson showed that the ankle moment contribution could reach as high as 45% during self-selected walking speeds in the OA group, compared to 26% in the control group. However, it is important to recognize the limitations of this study, including its cross-sectional design, which cannot establish causality, and the fact that the OA group had a significantly higher BMI and lower quadriceps strength, which are potential confounding factors.

Building on this, a recent 2-year longitudinal study by Chang and colleagues of 391 knees in 204 older adults (predominantly female) provides further insight into this compensatory redistribution of lower body kinetics during walking [118]. This study, which utilized both 3D gait analysis and MRI, found that smaller knee and larger ankle joint moment contributions are likely due to tibiofemoral structural degradation but may be protective of patellofemoral structural integrity. Conversely, larger knee and smaller ankle joint moment contributions may prevent further tibiofemoral damage at the expense of the patellofemoral joint. This longitudinal evidence is crucial as it suggests that these compensatory strategies have specific, and sometimes opposing, consequences for different compartments of the knee. This principle of kinetic redistribution is not limited to knee OA and older adults. It is also observed in middle-aged adults with meniscal degeneration, as shown by Holsgaard-Larsen and colleagues in a longitudinal study of 20 predominantly male patients undergoing arthroscopic partial meniscectomy [124]. Furthermore, it is a key feature in younger adults post-ACL reconstruction. The work of DeVita, Hortobagyi, and Barrier, though based on a very small sample of 8 patients, was seminal in demonstrating that even when kinematics appear normal after surgery, underlying kinetic deficits persist [120,121]. This highlights a critical concept: a “normal-looking” gait can mask significant biomechanical abnormalities. While these compensatory strategies are successful in preserving locomotion, they represent a fundamental disruption of the natural kinetic chain. By offloading the knee and increasing the mechanical demands on the hip and ankle, these adaptations may accelerate degeneration and lead to secondary pathologies in the compensating joints over time.

### 6.5. Muscle Weakness as a Driver of Cartilage Degeneration

A substantial body of evidence posits that quadricep weakness is a primary and modifiable risk factor for the development and progression of knee OA and associated cartilage loss [16,19,20,21,22,23,26,30,106,127,128]. As previously described, the knee extensors act as the knee’s primary dynamic stabilizer and shock absorber during locomotion. When these muscles are weak, they are less effective at counteracting ground reaction forces and controlling tibial translation, resulting in altered load distribution and increased stress on the articular cartilage [16,23,129]. Several meta-analyses have explored this relationship [21,130,131]. For example, Øiestad and colleagues found that quadriceps weakness in adults (N = 46,819) increases the odds of both radiographic (39% in men, OR 1.39; 43% in women, OR 1.85) and symptomatic knee OA (43% in men, OR 1.43; 85% in women, OR 1.85) [130]. However, the authors themselves rate the quality of this evidence as “low” due to significant heterogeneity between the included studies and the potential for publication bias. Similarly, Patterson and colleagues concluded that knee extensor weakness is associated with an increased risk (RR = Risk Ratio) of patellofemoral OA (RR 1.62) based on four studies and with tibiofemoral OA (RR 1.62) based on twelve studies, with follow-up periods ranging from 1 to 7 years. Furthermore, they found that quadricep weakness was associated with increased risk of cartilage lesions in the medial tibiofemoral (29%, RR 1.29) and lateral patellofemoral (41%, RR 1.41) compartments but not the lateral tibiofemoral or medial patellofemoral compartments. However, only 14 of 3737 identified studies were incorporated into their meta-analytic review. This relationship is further complicated by changes in muscle composition, such as an increase in fat infiltration (myosteatosis), which is associated with lower muscle strength, reduced power, and impaired functional mobility [26,29]. Myosteatosis, characterized by ectopic fat deposits in muscles (whether between muscles (intermuscular), within individual muscles (intramuscular), or inside muscle cells (intramyocellular) has been shown to impact mobility [132]. Consequently, these findings suggest that muscle dysfunction may precede and contribute to the development of degenerative joint changes, which will be further explored in the section on cartilage degradation as a driver of muscle dysfunction.

Mechanical instability caused by weak knee extensors triggers a common but counterproductive neuromuscular compensation: increased co-contraction of the hamstring muscles [133,134]. When the quadriceps cannot adequately stabilize the knee and absorb shock during locomotion, individuals with compromised knee health respond by excessively activating hamstring muscles during walking as a protective strategy [135,136,137]. While this coactivation represents the neuromuscular system’s attempt to provide stability that weakened quadriceps cannot deliver, it paradoxically may accelerate the very cartilage damage it seeks to prevent. Studies demonstrate that approximately 79.6% of ACLR patients exhibit increased co-contraction magnitude, with quadriceps strength consistently predicting muscle co-contraction levels; weaker individuals show greater compensatory activation [136,137]. However, this compensatory strategy proves biomechanically destructive. Research from Hubley-Kozey and colleagues reveals that muscle co-activity patterns provide critical information about the severity of knee OA, with lateral compartment differences progressing from asymptomatic to severe conditions and medial compartment differences distinguishing between osteoarthritic severities [138]. Thus, excessive co-contraction of the quadriceps and hamstrings dramatically increases compressive forces across the tibiofemoral and patellofemoral joints, creating abnormal loading patterns that accelerate cartilage breakdown beyond what weakness alone would cause [136].

The composition and morphology of the upper thigh muscles are important for the force-generating capacity of the knee joint to allow for proper mobility. Low muscle mass and strength are two of the three components of skeletal muscle function deficit [139]. Building on recommendations from the 2018 National Institute on Aging interdisciplinary workshop, myosteatosis in muscle health and aging was identified as the third component of skeletal muscle function deficit [132]. Research using animal models and in humans has observed increased inter- and intramuscular adipose tissue following injury [26,140]. Recently, Distefano and colleagues, using data from older adults enrolled in the Study of Muscle, Mobility, and Aging (age 76.1 ± 5 years, N = 655), observed that increased muscle fat infiltration was associated with greater knee OA severity, lower leg power, poorer mobility and reduced oxidative phosphorylation within the vastus lateralis via percutaneous biopsy [26]. These associations persisted after controlling for sex and adiposity, indicating that poor muscle composition due to muscle fat infiltration and lower leg power are linked inherently. While Distefano et al.’s observed a relationship between muscle fat infiltration and joint power, these associations range from weak to moderate (r = 0.21–0.41). Current mechanistic explanations, while biologically plausible, remain largely theoretical. One proposed mechanistic pathway points toward mitochondrial dysfunction [26,128]. Dysfunction in skeletal muscle mitochondria disrupts their ability to manage reactive oxygen species, which increased reactive oxygen species production is known to contribute to muscle atrophy and weakness [141]. However, direct evidence linking muscle composition to mitochondrial dysfunction remains an active area of research. Other possibilities lie within our genetics and muscle fiber typology. Downregulation of genes involved in mitochondrial oxidative metabolism and biogenesis, such as PPARα (peroxisome proliferator-activated receptor), PPARβ, or PGC-1α (peroxisome proliferator-activated receptor gamma coactivator 1-alpha) [142], and the shift in muscle fiber type from fast oxidative type IIa to less efficient hybridized type IIax fibers may contribute to reductions in lower leg power [25,29,143]. Despite this growing body of work suggesting fiber hybridization may play a role, its functional role in muscle composition and morphology remains inconclusive. Besides changes in muscle fiber typology, the extracellular matrix of skeletal muscle undergoes distinct changes with knee health. The extracellular matrix consists of three layers (basal lamina, perimysium, and epimysium) that play a vital role in the regulation and development of skeletal muscle [144]. Current evidence suggests that older adults with declining knee health may possess higher levels of extracellular matrix content compared to healthy controls, leading to reductions in muscle contractile force, increased profibrotic gene expression, and fewer satellite cells [25]. For a comprehensive review of the role of the extracellular matrix of muscle in OA pathogenesis, see the following reviews [144,145,146].

### 6.6. Cartilage Degradation as a Driver of Muscle Dysfunction

Conversely, pathology within the knee joint itself is a potent driver of muscle dysfunction. The primary mechanism is arthrogenic muscle inhibition (AMI), a continuous reflex inhibition of the musculature surrounding an injured or pathological joint [147,148]. AMI is triggered by altered afferent signals from mechanoreceptors in damaged joint structures and by the presence of joint effusion (swelling). Additionally, this neural inhibition prevents full voluntary activation of the knee extensors, leading to rapid and persistent weakness and atrophy. AMI is a frequently observed barrier in the rehabilitation of conditions such as ACL injury and knee OA, impeding recovery and threatening long-term health [147,148,149]. Furthermore, the process of cartilage degradation releases catabolic factors and pro-inflammatory cytokines into the joint environment, which can have a direct negative impact on the surrounding muscle tissue [150,151]. Age-related sarcopenia can lead to quadricep weakness, which increases mechanical stress on the cartilage, also initiating the degenerative process [152,153]. Once cartilage begins to degrade, it perpetuates muscle inhibition and weakness through AMI and inflammatory pathways. Consequently, this bidirectional, self-perpetuating cycle highlights the need for interventions that target both joint health and muscle function simultaneously.

The relationship between knee extensor muscle health and joint integrity can be conceptualized as a deviation-amplifying feedback loop [154], wherein small perturbations, such as muscle weakness or joint injury, initiate self-reinforcing chains of events that can drive either pathological degradation or adaptive remodeling. A vicious cycle may begin with an initial insult to the knee and subsequent AMI, leading to poor muscle activation, altered joint loading, and impaired shock absorption [148,149]. These biomechanical maladaptations accelerate cartilage degradation through the overexpression of pro-inflammatory cytokines [150], increased circulating reactive oxygen species [155], and the formation of disorganized extracellular matrix fragments that contribute to synovial inflammation and matrix breakdown [146,156]. Conversely, targeted exercise can initiate a virtuous cycle by improving muscle strength, modulating cytokine profiles through anti-inflammatory pathways, and promoting extracellular matrix clearance via enhanced matrix modeling [111,157]. Therefore, this adaptive loop can promote or degrade joint integrity depending on whether physiological changes scale toward macro-level biomechanical patterns.

The extracellular matrix of connective tissue also responds to mechanical loading and shares an intricate relationship with the extracellular matrix of skeletal muscle in maintaining knee health [146,156]. Ligaments, tendons, and cartilage are composed of specific signaling pathways that respond dynamically to mechanical loading and interact with the contractile elements in muscle to maintain knee health [146,156]. A decline in the mechanical properties or structural stability of non-muscular connective tissues can significantly impact joint biomechanics, resulting in irregular loading patterns [144,145,146]. Consequently, this declination may subsequently lead to muscle weakness and dysfunction either as a secondary effect or as part of a concurrent degenerative process. Growing evidence from research investigating cartilage biomarkers suggests a temporal precedence of cartilage degradation and inflammation. Several studies consistently observe that elevated levels of cartilage degradation markers, such as cartilage oligomeric matrix protein (COMP), crosslinked C-telopeptides of type II collagen (CTX-II), N-Terminal type 1 collagen telopeptide (NTX), and inflammatory cytokines like interleukin 6 (IL-6), interleukin 1β (IL-1β), and tissue necrosis factor alpha (TNF-α) are present in worsening knee health [150,151,158,159,160]. Although this is a growing area of research, this suggests that prominent features of knee health may reside within the cartilage and inflammatory pathways, with declining muscle health occurring as a subsequent or parallel event.

## 7. Discussion

The evidence synthesized in this review demonstrates that compromised knee health is a significant and growing global public health issue with serious societal and economic impacts that affect individuals throughout their healthspan. The knee is a complex, integrated system where structural integrity, neuromuscular function, and neural control are deeply interconnected to support the function and health of this joint. However, when knee joint health declines due to injury or age-related degeneration, a harmful chain reaction of biomechanical and physiological changes begins. This cycle involves muscle weakness, cartilage breakdown, and maladaptive neuromuscular adjustments, such as changes in lower extremity joint mechanics. While these adjustments may initially protect the joint, they eventually cause abnormal stress on the joint, faster structural damage, and reduced mobility. A key takeaway is the bidirectional, or mutually reinforcing, relationship between muscle health and joint integrity, where problems in one area directly cause issues in the other. This complex interaction highlights that effective treatments must target both the structural and neuromuscular aspects of knee health to break this vicious cycle, keep the joint healthy, and maintain mobility across the healthspan.

When knee health is compromised, a cascade of biomechanical compensation ensues to maintain function. Yet, there are several knowledge gaps in our understanding of this adaptive response that require further investigation. First, there is a need for early detection strategies, as notable discordance exists between structural damage and functional limitations. More sensitive physiological and functional biomarkers are required to identify at-risk individuals [150,161,162]. Second, the long-term consequences of compensation remain poorly understood; whether these strategies serve a protective or instead contribute to secondary or increased physiological strain is still unclear [17,163,164]. Finally, compensatory gait patterns are not metabolically neutral; they impose some physiological costs. A key gap remains in understanding how these biomechanical changes affect local muscle tissue function and metabolic health [50]. Bridging these gaps will advance both preventative and rehabilitative approaches in clinical biomechanics.

### 7.1. The Integrated Knee Health-Mobility Model (IKHMM): A Dynamic Theoretical Framework

To address these gaps, we propose the Integrated Knee-Health-Mobility Model (IKHMM) as a comprehensive framework for understanding the dynamic, bidirectional relationships between knee joint integrity, muscle function, biomechanical adaptations, and mobility outcomes across the healthspan. The IKHMM conceptual model moves beyond a singular focus on structural disease by synthesizing key domains that collectively determine a person’s mobility, as shown in Figure 6. Specifically, it integrates (1) structural integrity, such as radiographic changes and tissue health; (2) neuromuscular function, including quadriceps strength, proprioception, and muscle quality; (3) biomechanical adaptation, encompassing gait mechanics and joint loading patterns; and (4) functional mobility. At its core, the model posits that it is the dynamic interplay between these domains, rather than any single factor in isolation, that dictates the ultimate mobility outcome, offering a more holistic foundation for developing targeted and personalized rehabilitation strategies, illustrated in Figure 5.

#### 7.1.1. Domain 1: Structural Integrity

The structural integrity of the knee joint encompasses the physical health of its constituent tissues, including articular cartilage, subchondral bone, menisci, ligaments, and synovial tissue. A comprehensive assessment necessitates a multimodal approach for quantifying knee joint health [32]. While weight-bearing radiographs, employing the Kellgren-Lawrence grading system, offer cost-effectiveness, their sensitivity to early changes is limited, and they exhibit some discordance with clinical symptoms [32,165,166]. Quantitative magnetic resonance imaging demonstrates greater sensitivity to incipient abnormalities within cartilage, bone marrow, menisci, and synovium [32,167]. Alternatively, ultrasonography presents a safe and economical option for visualizing alterations in knee structures, albeit requiring a proficient operator [168,169]. The refinement and development of artificial intelligence and machine learning algorithms to improve classification using these diverse imaging technologies is also pertinent. From a physiological perspective, the identification of a robust set of biomarkers definitively associated with symptomatology and structural abnormalities remains necessary [146,156]. However, imaging findings alone may not fully capture the clinical picture. Studies show that radiographic severity correlates with patient-reported symptoms in only about 50% of cases [161], underscoring the need to integrate clinical assessments, such as pain, swelling, range of motion, and joint stability, for a more comprehensive diagnostic and prognostic perspective.

#### 7.1.2. Domain 2: Neuromuscular Function

The neuromuscular domain focuses on the health and performance capabilities of the muscles surrounding the knee, including muscle composition, metabolic capacity, force generation, neuromuscular control, and intermuscular coordination [15,19,20,29,30,135,170,171,172,173,174,175]. Several studies from the 1990s identified quadriceps weakness as a predictor of knee OA [12,22,67,111,121,171,176,177]. Animal models have further demonstrated that inducing quadriceps weakness can lead to the degeneration of articular surfaces [19,30,178]. Growing evidence also indicates that fat infiltration in the vastus lateralis is indicative of deteriorating knee health and is associated with lower strength, reduced power, and poorer functional mobility [25,26,29,143]. Further research is required to elucidate the role of muscle weakness in structural degradation and the consequent increase in metabolic cost during walking.

#### 7.1.3. Domain 3: Biomechanical Adaptation

The biomechanical adaptation domain describes how mechanical forces are transmitted through the knee and how movement patterns adjust in response. This encompasses joint kinetics, kinematics, load distribution, and compensatory strategies. As discussed previously, a consistent finding across acute and chronic knee pathologies is the redistribution of joint kinetics to offload the affected knee [118,120,121,123,124,125,126]. A concerted effort to standardize walking paradigms for assessing biomechanical adaptations remains necessary.

#### 7.1.4. Domain 4: Functional Mobility

The mobility domain represents the ultimate expression of the knee system’s overall health, encompassing gait performance, functional capacity, metabolic efficiency, and participation in physical activities. Slower walking speed, shorter stride length, and lower cadence are often associated with declining knee health [91,162]. However, among these outcomes, gait speed has stood out as a straightforward and reliable indicator of declining knee health, regardless of whether it is due to symptoms or radiographic findings. Data from the OA Initiative shows that adults with symptomatic OA are nine times more likely to experience reductions in gait speed than those with only radiographic evidence [179]. Furthermore, a decline in gait speed is strongly associated with increased bone marrow lesion volume and effusion, even a year before disease onset [180]. Given that a gait speed below 0.8 m/s is a recognized threshold for increased mortality risk in older adults [181,182], these findings underscore the clinical significance of gait speed as a key indicator of worsening knee health.

#### 7.1.5. Feedback Mechanisms

The IKHMM acknowledges the role of feedback loops in either enabling or mitigating knee dysfunction. First, Positive Feedback Loops amplify joint deterioration. This feedback loop is based on the Protective Response theory discussed in Merkle, Sluka, and Frey-Law’s clinical commentary on the interaction between pain and movement that draws on extensive work by Hodges [183,184]. The central tenet of this theory is that injury, pain, or the threat of either can invoke a wide range of motor behavior responses [184]. For instance, pain leads to arthrogenic muscle inhibition, causing weakness, which in turn results in altered mechanics and increased stress, ultimately leading to further pain. Negative feedback refers to the adaptive compensation that involves dysfunction, which prompts motor adaptation and load redistribution, thereby stabilizing the system.

In addition to acknowledging the significance of feedback mechanisms in musculoskeletal decline, we propose a set of interconnected propositions with the IKHMM that warrant further scholarly investigation. The first proposition, *cascading deterioration*, posits that knee dysfunction rarely occurs in isolation. An initial insult (whether structural, neuromuscular, or biomechanical) can initiate a cascade of compensatory responses that, while initially adaptive, may accelerate systemic deterioration. For instance, structural damage may induce neuromuscular reorganization, thereby altering joint loading patterns and ultimately leading to diminished mobility and further decline. The second proposition, *adaptive compensation*, underscores the body’s capacity to maintain function despite localized impairment. When knee mechanics are compromised, the locomotor system redistributes mechanical demands among adjacent joints and muscle groups. This redistribution may contribute to the preservation of mobility; however, it often results in increased energy expenditure and long-term inefficiency. Another proposition, *Metabolic–Mechanical Coupling*, suggests that local muscle metabolic capacity influences overall movement quality. Declines in oxidative efficiency reduce endurance, modify movement strategies, elevate the energy cost of fundamental tasks, and contribute to fatigue-related limitations. The final preposition, the *Threshold Effect*, offers a plausible explanation for the observed dissociation between structural damage and symptom severity. Functional decline can remain latent until cumulative impairments surpass a critical threshold, at which point compensatory mechanisms fail, and mobility declines precipitously.

#### 7.1.6. Clinical Implications

The IKHMM provides a comprehensive framework for clinical practice, advocating a shift beyond traditional imaging to adopt multimodal assessment approaches. These include detailed structural assessments using advanced imaging, comprehensive functional evaluations of muscle capacity and control, and in-depth biomechanical analyses of movement patterns. The IKHMM also guides tiered intervention strategies: primary prevention optimizes neuromuscular function, secondary prevention enhances adaptive compensation to slow decline, and tertiary prevention focuses on managing advanced stages to preserve function. Additionally, the model supports precision medicine, enabling personalized risk profiling through genetic, biomechanical, and metabolic insights to tailor interventions effectively.

#### 7.1.7. Research Applications

Beyond clinical applications, the IKHMM could serve as a powerful guide for research initiatives. It facilitates hypothesis generation, formulating testable hypotheses regarding the temporal sequences of decline across different domains, the effectiveness of various domain-specific interventions, the role of individual variation in adaptive capacity, and the relationships between biomarkers and system components. For study design implications, the model encourages the adoption of longitudinal designs to capture dynamic interactions over time and multimodal measurement approaches to comprehensively assess multiple domains. It also promotes research focused on optimizing the timing of interventions for maximum impact. In terms of translation pathways, the IKHMM can guide the development of algorithms for early detection, personalized intervention protocols, and risk stratification tools.

#### 7.1.8. Limitations

This review, while drawing from multiple databases with relevant terminology, did not adhere to systematic review methodology, lacking predetermined inclusion/exclusion criteria, risk of bias assessment, and quality appraisal tools. The selection of studies may exhibit authorial bias toward specific research areas or methodological approaches, potentially circumscribing the comprehensiveness of evidence synthesis. Furthermore, the primary focus was on ambulatory mobility, thereby underemphasizing other pertinent activities such as stair climbing, sit-to-stand transitions, running, and jumping. These limitations underscore several critical imperatives for future research: (1) the execution of systematic reviews employing rigorous methodology to explore specific facets of knee health-mobility relationships; (2) the initiation of large-scale, diverse, longitudinal cohort studies featuring standardized outcome measures and extended follow-up durations; (3) the conduct of mechanistic studies with adequate sample sizes, diverse populations, and comprehensive biomarker assessment; and (4) the implementation of research examining the real-world translation of knee health assessment and intervention strategies. Notwithstanding these limitations, convergent evidence across various domains provides a robust foundation for conceptualizing knee health as a complex and dynamic phenomenon that influences mobility throughout the lifespan. Subsequent research addressing these limitations will strengthen the evidence base, improve external validity, and facilitate translation to diverse clinical and community settings.

## 8. Conclusions

The evidence covered within this review substantively advocates for a paradigmatic shift in the conceptualization of knee health, transitioning from a reactive approach targeting established pathology to a proactive, lifespan-oriented management strategy. The proposed Integrated Knee Health-Mobility Model provides a comprehensive framework for elucidating the complex interactions that underpin the preservation of mobility. Furthermore, it serves as a foundation for devising more efficacious prevention strategies, earlier intervention protocols, and individualized treatment plans. The success of this paradigm will depend on sustained interdisciplinary collaboration, technological advancements, and a long-term research commitment that captures the intricate dynamics of knee health across the healthspan.

## Figures and Tables

**Figure 1 jfmk-10-00454-f001:**
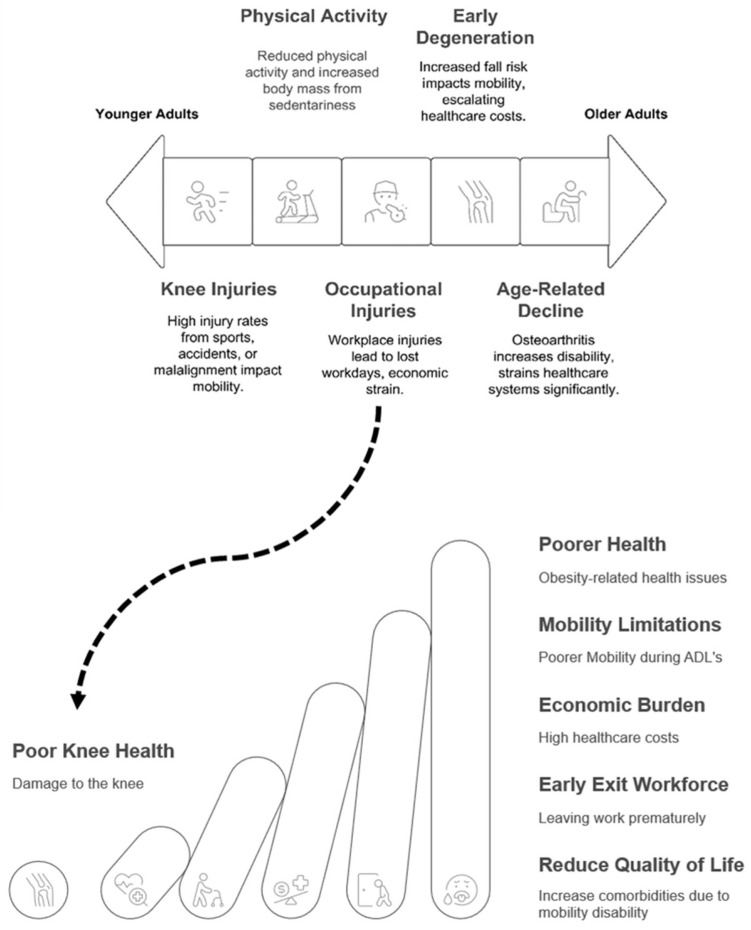
Compromised Knee health impacts all ages across the healthspan. Knee health deteriorates across the lifespan through multiple pathways. For many, this process begins with sports injuries and occupational hazards in younger adults, progressing through early degenerative changes in middle age, and culminating in age-related decline in older adults. Poor knee health creates a cascading effect (depicted by the curved arrow) that progressively worsens mobility limitations, increases the economic burden through elevated healthcare costs, forces early workforce exit, and ultimately reduces quality of life through increased comorbidities and mobility disability. The ascending pillars illustrate the increasing severity of these consequences over time, highlighting how knee health plays a critical role in determining functional independence and overall healthspan throughout the aging process.

**Figure 2 jfmk-10-00454-f002:**
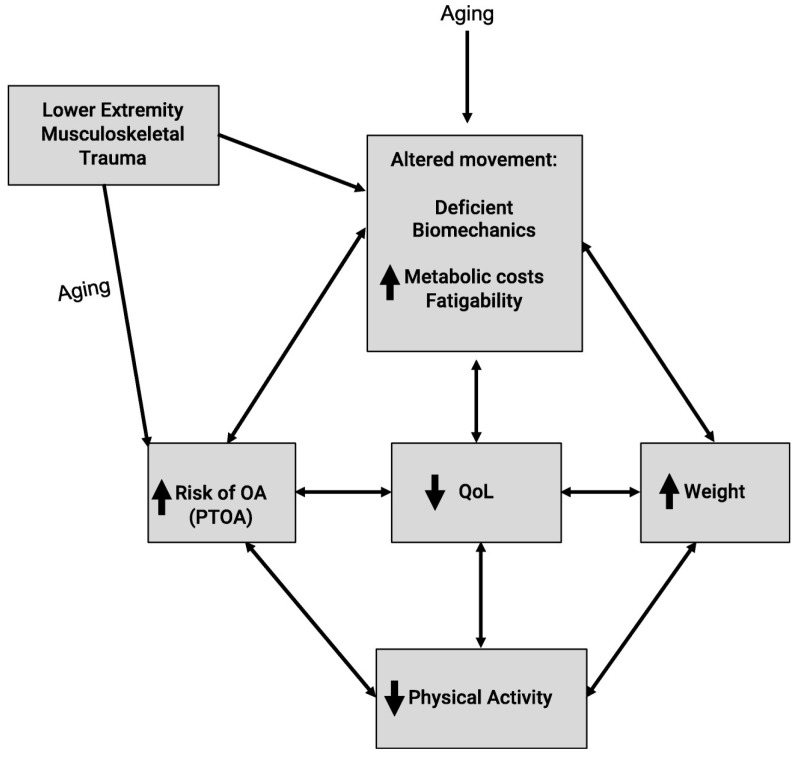
Conceptual flow chart demonstrating how lower extremity musculoskeletal trauma can incur greater metabolic costs. Reproduced with permission from Landers-Ramos & Custer [50], Current Geriatric Reports. © 2021 Springer Nature. Permission granted by the publisher.

**Figure 3 jfmk-10-00454-f003:**
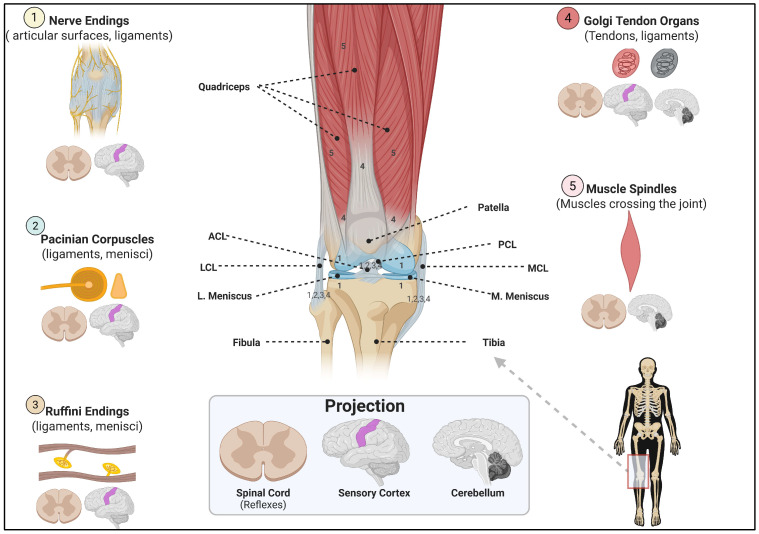
Structure and Mechanoreceptors of the Knee: The localization, functional roles, and central projections of key mechanoreceptors embedded within both passive and dynamic knee structures. Free nerve endings, which are unmyelinated, relay nociceptive signals from multiple joint tissues to the dorsal horn of the spinal cord and sensory cortex. Pacinian corpuscles, which are fast-adapting and pressure-sensitive, are situated in the joint capsules, menisci, and ligaments, transmitting high-frequency vibration and pressure signals via rapidly conducting fibers to the spinal cord and sensory cortex. Ruffini endings, which are slow-adapting and sensitive to stretch, reside in the menisci and ligaments and project low-threshold afferent fibers with projections in the spinal cord and sensory cortex. Golgi tendon organs located in muscle tendons and ligaments respond to force and tension and transmit proprioceptive feedback through afferent fibers. These mechanoreceptors project through spinal afferent pathways to both spinal and supraspinal centers, including the cerebellum (muscles) and the sensorimotor cortex (capsule and menisci), contributing to reflexive and conscious proprioceptive control, which is essential for joint stability and coordinated movement. Muscle spindles respond to elongation, velocity, and acceleration in muscles that cross the knee joint and connect to the spinal cord and cerebellum. ACL = Anterior Cruciate Ligament, MCL = Medial Collateral Ligament, M. Meniscus = Medial Meniscus, LCL = Lateral Collateral Ligament, L. Meniscus = Lateral Meniscus, PCL = Posterior Cruciate Ligament. This figure was created in BioRender.

**Figure 4 jfmk-10-00454-f004:**
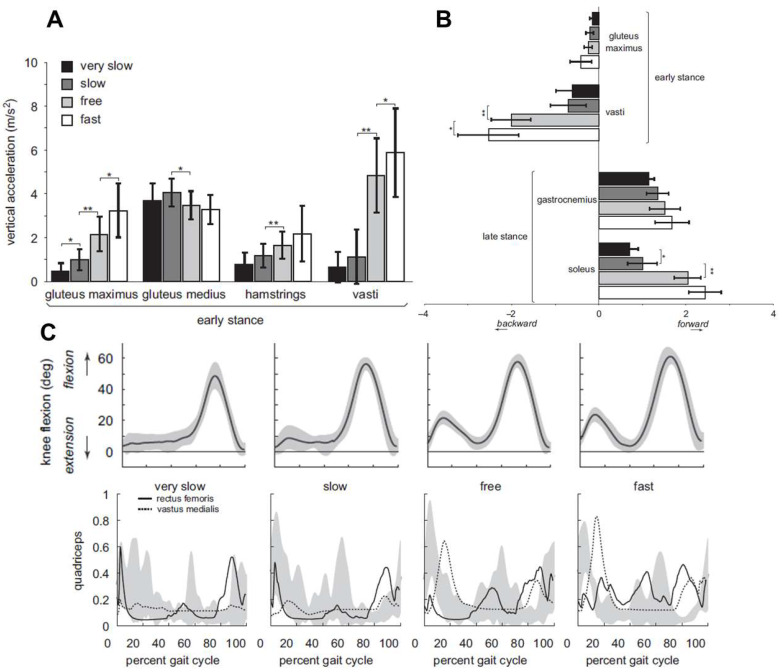
Vasti muscle activation and knee flexion across walking speeds. (**A**) Vertical acceleration profiles (m/s^2^) highlight increased engagement of the vasti group (primarily vastus lateralis and medialis) with progressive walking speeds, with significant inter-speed differences marked by asterisks. (**B**) Vasti activation patterns dominate during early stance and backward movement phases, showing distinct modulation from slow to fast conditions. (**C**) Knee flexion angle trajectories over the gait cycle (0–100%) align with elevated and earlier vasti activation, suggesting their role in stabilizing the knee as walking speed increases. * *p* < 0.05 and ** *p* < 0.01 indicates significance for within-subject repeated contrast analyses. Reproduced with permission from Liu et al. [77], Journal of Biomechanics. © 2008 Elsevier. Permission granted by the publisher.

**Figure 5 jfmk-10-00454-f005:**
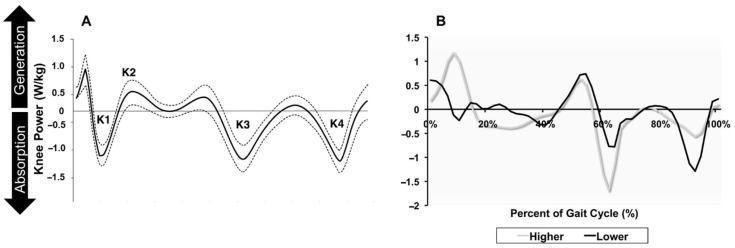
Normal and Pathological Alterations in Knee Power during Walking. (**Panel A**) illustrates typical knee joint power (W/kg) across the gait cycle in healthy adults, with distinct phases (K1–K4) marked to indicate periods of power absorption and generation. The solid line represents average knee power, with dashed lines denoting variability around the mean. (**Panel B**) contrasts knee power profiles in high-functioning (grey line) and low-functioning (black line) adults with knee osteoarthritis. Differences in knee power generation and absorption reflect functional alterations throughout the gait cycle. (**Panel B**) From Segal et al. [116]. The grey line indicates a higher functioning adult with knee OA. The black line indicates a lower functioning adult with knee OA. (**Panel B**) reproduced with permission from Segal et al. [116], Archives of Physical Medicine and Rehabilitation. © 2009 Elsevier. Permission granted by the publisher.

**Figure 6 jfmk-10-00454-f006:**
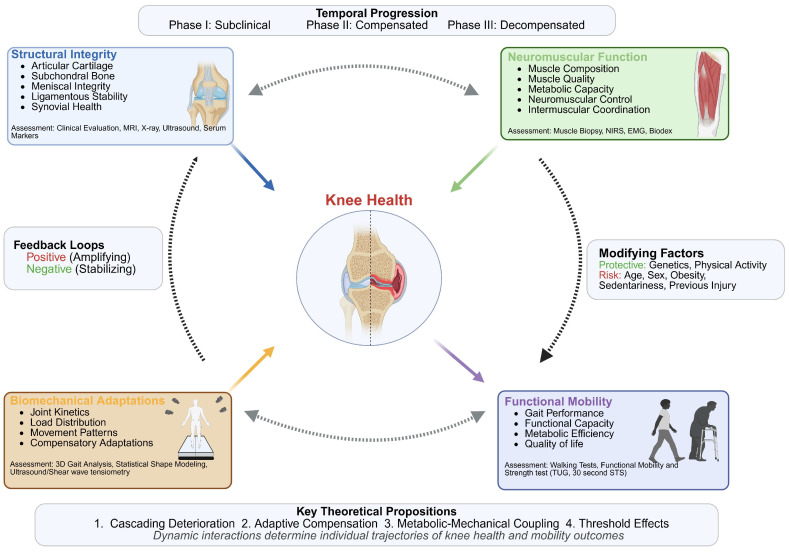
The Integrated Knee Health-Mobility Model (IKHMM): A Comprehensive Theoretical Framework for Understanding Dynamic Knee Health Across the Healthspan. The IKHMM conceptualizes knee health as a complex, adaptive system comprising four interconnected domains positioned around a central knee joint illustration. Structural Integrity (blue, **top left**) encompasses articular cartilage, subchondral bone, meniscal integrity, ligamentous stability, and synovial health, assessed through MRI, X-ray, ultrasound, and serum markers. Neuromuscular Function (green, **top right**) includes muscle composition, muscle quality, metabolic capacity, neuromuscular control, and intermuscular coordination, evaluated via muscle biopsy, NIRS, EMG, and Biodex testing. Biomechanical Adaptations (orange, **bottom left**) covers joint kinetics, load distribution, movement patterns, and compensatory strategies, measured through 3D gait analysis, ultrasound, and shear wave tensiometry. Functional Mobility (purple, **bottom right**) encompasses gait performance, functional capacity, metabolic efficiency, and quality of life, assessed using walk tests, TUG, 30 s STS, and other functional measures. The Temporal Progression header displays three distinct phases: Phase 1 (Subclinical), Phase 2 (Compensated), and Phase 3 (Decompensated), representing the evolution of knee health decline over time. Bidirectional arrows connecting each domain to the central knee joint illustrate dynamic interactions, while Feedback Loops are distinguished by color—positive loops (amplifying dysfunction) and negative loops (stabilizing mechanisms). Modifying Factors (gray box, **right**) highlight protective elements (genetics, fitness) and risk factors (age, obesity, injury) that influence individual trajectories. The Key Theoretical Propositions at the bottom outline four core mechanisms: (1) Cascading Deterioration—dysfunction in one domain triggering compensatory responses in others; (2) Adaptive Compensation—the system’s capacity to redistribute demands maintaining function; (3) Metabolic-Mechanical Coupling—relationships between local muscle efficiency and global mobility performance; and (4) Threshold Effects—critical points where compensatory capacity is exceeded, leading to rapid functional decline. This framework guides research investigations into knee health mechanisms and clinical interventions targeting multiple domains to preserve mobility across the healthspan. Abbreviations: MRI = Magnetic Resonance Imaging, NIRS = Near-infrared Spectroscopy, EMG = Electromyography, TUG = Timed Up and Go, 30 s STS = 30 s Sit-to-Stand.

## Data Availability

No new data were or analyzed in this study. Data sharing is not applicable to this article.

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
