# Peer review of "J. Funct. Morphol. Kinesiol.2025, 10(4), 454;https://doi.org/10.3390/jfmk10040454"

_jfmk, 2025, doi:10.3390/jfmk10040454_

Round 1

Reviewer 1 Report

Comments and Suggestions for Authors

Dear Authors,

Thank you for the opportunity to review your manuscript entitled “Knee Health Governs Mobility Across the Healthspan.” Your study presents a comprehensive and well-structured narrative review addressing the multifaceted relationship between knee health and mobility throughout life. The proposed Integrated Knee Health-Mobility Model (IKHMM) is conceptually strong and offers a valuable theoretical framework integrating structural, neuromuscular, biomechanical, and functional domains. The topic is both timely and relevant to current scientific and clinical needs.
However, several methodological and conceptual aspects require clarification and refinement to improve scientific rigor and overall clarity.

- Major Comments

  • Page 1, lines 23-44: Although not specific to this section, the manuscript would benefit from adding a new section titled “Personal and Lifestyle Determinants of Knee Health.” This could become Sections 2 and 3 of the paper. In this section, please address key personal determinants such as sex, age, body mass index, genetics, sports participation history, past injuries, muscular deficiencies, limb asymmetry, lower-limb dominance, sedentary behavior, and comorbidities. While some of these factors are discussed throughout the text, they appear scattered and fragmented, making it difficult to synthesize and interpret their collective impact. Consolidating them into a dedicated section will improve coherence and strengthen the foundation of the IKHMM framework. This addition should also be reflected in Figures 1 and 6 as “Personal and Lifestyle” and/or “Modifying Factors.”
  • Page 2, lines 60-69: It is recommended to add a subsection titled “Methods.” Although a fully systematic approach may not be required for this study’s type, the current text already includes some information about study selection. Expanding this section will enhance transparency and reduce potential selection bias. Please provide details such as: inclusion and exclusion criteria; study types selected; a clear definition of what constitutes “high-quality” and “robust methodologies”; whether the search was performed by title, title/abstract, or full text; and a concise explanation of the screening and selection process. Including these details would substantially improve transparency and reproducibility.
  • Page 3, lines 101-120: The section discussing the economic burden of poor knee health should be more explicit in distinguishing direct and indirect costs. Consider citing at least one study that quantifies both types of costs (e.g., direct cost of knee injury was X, with indirect cost of Y). As currently written, the association between total costs and specific contributors is unclear.
  • Page 3, line 137: The expression overuse of knee injuries should be revised. The traditional concept of “wear and tear” is outdated; current mechanobiological evidence shows that the concept of “overuse” has often been misinterpreted, as regular and controlled mechanical loading is not inherently harmful and is, in fact, essential for maintaining tissue health and adapted. The issue arises when the loading becomes excessive or insufficiently controlled, which may lead to structural stress and injury. Therefore, it is suggested that the authors clarify the terminology used in this context to better reflect the distinction between beneficial adaptive loading and pathological excessive loading. Therefore, misuse or disuse are more harmful than the “overuse”. So, it is advised to review and update this terminology and the associated ideas/theories.
  • Page 3, lines 158-201: This section should also address limb length asymmetry and its significance for knee health. Additionally, please discuss biomechanical sex differences, emphasizing disparities in muscle strength and alignment (e.g., valgus angle).
  • Page 7, lines 234-236: The authors focus primarily on flexion-extension control (walking and running). However, rotational and pivot movements are critical for dynamic stability and injury prevention. Please add a brief paragraph acknowledging that rotational and pivot motions are essential degrees of freedom influencing ligament strain and proprioceptive control, especially during cutting or change-of-direction tasks.
  • Page 7, lines 243-245: The sensorimotor system listed the menisci as key proprioceptive structures. This is not entirely accurate. Menisci play a secondary role compared to ligaments, tendons, and muscles. Please revise this statement by removing the menisci from the list of primary proprioceptive structures.
  • Page 15, lines 582-596: Although the section on structural integrity covers key imaging modalities, clinical assessment parameters (pain, swelling, range of motion, joint stability, and palpation findings) should also be included. Clinical evaluation remains essential, as imaging findings alone may not correlate with symptoms (e.g., in knee OA, radiographic findings align with clinical symptoms in only about 50% of cases). Incorporating clinical and functional assessments will provide a more complete diagnostic and prognostic perspective. These aspects should also be represented in Figure 6.

- Minor Comments

  • Page 1, lines 32-34: Missing space between “time” and “[9].”
  • Page 1, lines 38-40: Too many commas; please revise the sentence for clarity.
  • Page 2, lines 43-44: The word “synonym” should be reviewed for appropriateness.
  • Page 2, lines 45-59: These two paragraphs should be merged.
  • Page 2, lines 73-75: “Osteoarthritis” was already abbreviated (page 1, line 39); use only “OA.”
  • Page 2, lines 80-83: “Anterior cruciate ligament” was already abbreviated (page 1, line 39); use only “ACL.”
  • Page 3, lines 92-94: Use “OA” instead of the full term.
  • Page 3, lines 127-130: Use “OA” instead of the full term.
  • Page 3, lines 134-137: Spell out “United States” before later abbreviating as “U.S.”
  • Page 5, lines 191-194: Use “ACL” instead of the full term.
  • Page 5, lines 196-199: Missing space between “degeneration” and “[52].”
  • Page 5, lines 199-201: Missing space between “walking” and “[11].”
  • Page 7, lines 231-233: Missing space between “(LCL)” and “[77].”
  • Page 7, lines 243-246: Missing space between “proprioception” and “[82, 83].”
  • Page 8, line 279: Missing space between “=” and “Posterior.”
  • Page 10, lines 239-240: Missing space between “muscles” and “[59].”
  • Page 10, lines 346-351: Missing space between “(see Figure 5A)” and “[57].”
  • Page 11, lines 372-376: Missing space between “(the sum of moments from the ankle, knee, and hip)” and “[75].”
  • Page 12, lines 413-416: Excessive spacing between “39%” and “in.”
  • Page 13, lines 445-449: Use “OA” instead of the full term.
  • Page 13, lines 467-469: Missing space between “dysfunction” and “[26, 102].”
  • Page 13, lines 469-472: Missing space between “weakness” and “[115].”
  • Page 13, lines 487-489: Use “OA” instead of the full term; also, in the end of the sentence is “see [118-120]” – appears incomplete; please check for missing text.
  • Page 13, lines 491-493: Missing space between “joint” and “[121, 122].”
  • Page 14, lines 512-516: Missing spaces between “cytokines” and “[124],” “species” and “[129],” and “breakdown” and “[120, 130].”
  • Page 14, lines 516-519: Missing space between “modeling” and “[86, 131].”
  • Page 14, lines 522-524: Excessive spacing between “[120, 130].” and “Ligaments.”
  • Page 14, lines 526-528: Missing space between “patterns” and “[118-120].”
  • Page 16, lines 593-596: Misplaced “=” symbol.
  • Page 16, lines 598-601: Missing space between “coordination” and “[15, 19, 20, 29, 30, 109, 600 144-149].”
  • Page 18, lines 661-661: Excessive space between “acknowledges” and “the.”

Author Response

Thank you very much for taking the time to thoroughly review our manuscript. We appreciate your feedback and have included your suggestions in the latest revision. Please find the detailed responses below, along with the corresponding revisions/corrections highlighted in track changes in the resubmitted files.

Comments 1:
Page 1, lines 23-44: Although not specific to this section, the manuscript would benefit from adding a new section titled “Personal and Lifestyle Determinants of Knee Health.” This could become Sections 2 and 3 of the paper. In this section, please address key personal determinants such as sex, age, body mass index, genetics, sports participation history, past injuries, muscular deficiencies, limb asymmetry, lower-limb dominance, sedentary behavior, and comorbidities. While some of these factors are discussed throughout the text, they appear scattered and fragmented, making it difficult to synthesize and interpret their collective impact. Consolidating them into a dedicated section will improve coherence and strengthen the foundation of the IKHMM framework. This addition should also be reflected in Figures 1 and 6 as “Personal and Lifestyle” and/or “Modifying Factors.

Response 1: We appreciate this excellent suggestion and agree with this comment. Therefore, we have added a section titled “Personal and Lifestyle Determinants of Knee Health” with the subheadings about aging and injury, obesity, and sex and genetic factors. Additionally, we cite relevant reviews in case the readers would like more information about a specific factor. Pages 6 and 7, lines 151-199, contain the updated text. We have also updated Figures 1 and 6 with the new information. The new sections are as follows:

Personal and Lifestyle Determinants of Knee Health

Age and Injury History

As previously mentioned, age is a significant factor in knee health, with multiple risks manifesting throughout the lifespan. Epidemiological evidence suggests that sports-related knee injuries sustained during young adulthood are as-sociated with an increased likelihood of developing radiographically confirmed knee osteoarthritis (OA) in later life, with signs of OA detectable within one year of ACL reconstruction [56]. Although ACL reconstruction is associated with high rates of return to sport participation [41], it concurrently correlates with a 4.7-fold increase in the risk of developing moderate to severe knee OA [57]. In addition to sports-related knee injuries, any injury to the knee—including ligament, tendon, meniscus injuries, or fractures—results in approximately a sixfold increase in the risk of developing knee OA [58]. This risk is further amplified by the malalignment of the knee's mechanical axis, commonly observed in conditions such as genu varum and valgus deformities [59]. While diagnosis for declining knee health may peak during late mid-dle adulthood (age 55 to 65) [60], the path may be set decades prior by injury or malalignment. Taken together, these findings highlight the importance of maintaining knee health early in life to prevent long-term degeneration.

Obesity

Excess body mass is a significant and modifiable risk factor for knee health [61]. Elevated body mass index (BMI), particularly when coupled with lower limb malalignment, may increase mechanical loading on the tibiofemoral and patellofemoral joints, accelerating cartilage deterioration [62]. Besides mechanical influences, obesity is characterized by a state of chronic, low-grade systemic inflammation [61,63]. Adipose tissue in obese individuals secretes pro-inflammatory cytokines and adipokines that can contribute to cartilage degradation [61]. While several studies have shown that increased body mass is associated with increased risk of knee OA [61,64], the underlying mechanisms re-main multifactorial and complex [65]. Abnormal mechanical loading, obesity-related malalignment, and muscle weak-ness may disrupt cartilage homeostasis and bone remodeling factors [62,66]. For a comprehensive review of obesi-ty-related knee OA, see Chen et al.'s review on this topic [66].

Sex and Genetic Factors

Sex is a strong factor in declining knee health. Several studies have shown that females have a higher prevalence and more symptoms of poorer knee health than males, due to a mix of hormonal, anatomical, and biomechanical dif-ferences [67-69]. These include reduced cartilage volume, different joint loading patterns, and diverse inflammatory pro-files [68]. Furthermore, distinct anatomical and biomechanical factors include a larger quadriceps (Q) angle (the angle formed by the pull from the anterior superior iliac spine to the center of the patella) and reduced muscle strength [68]. Larger static Q-angles are more frequently observed in women because of a wider pelvis and distinctive femoral and hip structures [70]. These anatomical features are associated with changes in patellar tracking and increase the risk of pa-tellofemoral pain and dysfunction [59,68,71]. Besides the anatomical differences, hormonal changes during menstrua-tion in female athletes and following menopause in older women can impact knee health [72,73]. For a comprehensive review of sex-specific factors associated with declining knee health, see Segal et al.'s review on this topic [68].

Genetic predisposition is a material contributor to knee OA risk. Recently, a large-scale Genome-Wide Association Study (GWAS) using UK Biobank and multi-omics data from nearly 140,000 participants with (~23,000 cases of radio-graphically confirmed and self-report hip and knee OA) identified nine new OA loci [74]. Zengini et al.'s findings demonstrated that hip and knee OA genetic expression profiles were highly correlated but distinct depending on joint-specific traits such as joint space width. For instance, they found that the single-nucleotide polymorphism (a varia-tion at a single position in a DNA sequence) rs11780978 on chromosome 8 near the Plectin gene (PLEC) is associated with joint space narrowing [74,75]. Beyond these genetic variations, they established a causal relationship between high BMI, lumbar spine bone mineral density, and knee OA. Additional analyses revealed that common genetic variants ex-plained about 15% of knee OA heritability, suggesting they may play a role in OA development. Please see Zengini et al.’s and Sorial et al.’s research for more details on the epigenetic variants that are associated with knee OA [74,75].”

Comments 2: Page 2, lines 60-69: It is recommended to add a subsection titled “Methods.” Although a fully systematic approach may not be required for this study’s type, the current text already includes some information about study selection. Expanding this section will enhance transparency and reduce potential selection bias. Please provide details such as: inclusion and exclusion criteria; study types selected; a clear definition of what constitutes “high-quality” and “robust methodologies”; whether the search was performed by title, title/abstract, or full text; and a concise explanation of the screening and selection process. Including these details would substantially improve transparency and reproducibility.

Response 2:  Thank you for this great suggestion. Accordingly, we have incorporated a methods section for transparency and reproducibility. The additional section can be found on Page 2, lines 57 – 77, with the following text:

“Methods

This narrative review was organized using a systematic search method to offer a thorough synthesis of the litera-ture. We conducted searches on PubMed, MEDLINE, SPORTDiscus, and Google Scholar for peer-reviewed studies pub-lished from 1980 to 2025. Additionally, the Research Rabbit AI tool was employed to visualize connections between pa-pers and pinpoint key articles. The search combined keywords and MeSH terms such as "knee osteoarthritis," “Knee OA,” “symptomatic knee OA,” ”radiographic knee OA,” "anterior cruciate ligament," "ACL," "meniscus," "knee injury," "quadricep weakness," "arthrogenic muscle inhibition," “healthcare costs,” “sensorimotor system,” “proprioception,” "muscle function," “biomechanics”, "gait," "joint power," “joint moment,” "total support moment," "mobility," "locomo-tion," and "walking."

The search mainly targeted titles and abstracts to identify relevant articles, followed by a full-text review to assess eligibility and synthesize findings. The inclusion criteria included peer-reviewed articles in English that explored the relationship between core concepts in adult human populations. Animal studies were only incorporated to provide mechanistic insights and were not the primary focus. Exclusion criteria included case reports, dissertations, conference abstracts, and non-peer-reviewed sources. To improve the reliability of the review, we emphasized "high-quality studies" and "robust methodologies," such as systematic reviews, meta-analyses, large-scale longitudinal cohort studies (like the Osteoarthritis Initiative), and key mechanistic studies that establish foundational knowledge. Although this was not a formal systematic review, this approach aimed to minimize selection bias and develop a comprehensive, evi-dence-based foundation for the synthesis and the proposed theoretical model.”

Comment 3: Page 3, lines 101-120: The section discussing the economic burden of poor knee health should be more explicit in distinguishing direct and indirect costs. Consider citing at least one study that quantifies both types of costs (e.g., direct cost of knee injury was X, with indirect cost of Y). As currently written, the association between total costs and specific contributors is unclear.

Thank you for pointing this out. We agree that the direct and indirect costs are unclear in this section. Accordingly, we have added the following statements to provide insight into the economic impact of OA.

“Direct costs, which include medical expenses for treatment and healthcare utilization, are substantial. Although not specific to poor knee health, there are significantly higher annual OA-related healthcare costs. Among insured adults, OA increased direct insurer expenses by $4,833 for women and $4,036 for men, with out-of-pocket costs rising by $1,379 and $694, respectively [46]. Nationwide, this resulted in an estimated $185.5 billion in yearly OA-related spending, with women contributing a larger share due to higher prevalence and per capita costs. Indirect costs, which represent lost economic output from disability and reduced productivity, are equally impactful. Knee injuries represent a leading cause of lost workdays [45]. Isaacson et al.'s longitudinal analysis of military health records (2016–2021) revealed that knee injuries were the most common lower extremity musculoskeletal injury, resulting in over half a billion dollars in costs, a figure reflecting both treatment and lost duty time [43].”

Comment 4: Page 3, line 137: The expression overuse of knee injuries should be revised. The traditional concept of “wear and tear” is outdated; current mechanobiological evidence shows that the concept of “overuse” has often been misinterpreted, as regular and controlled mechanical loading is not inherently harmful and is, in fact, essential for maintaining tissue health and adapted. The issue arises when the loading becomes excessive or insufficiently controlled, which may lead to structural stress and injury. Therefore, it is suggested that the authors clarify the terminology used in this context to better reflect the distinction between beneficial adaptive loading and pathological excessive loading. Therefore, misuse or disuse are more harmful than the “overuse”. So, it is advised to review and update this terminology and the associated ideas/theories.

Thank you for this great comment. We agree that overuse is a vague way to describe the maladaptive response to excessive mechanical stress or joint instability. We have removed every instance of “overuse” and replaced it with “excessive mechanical stress or joint instability” where possible.

Comment 5: Page 3, lines 158-201: This section should also address limb length asymmetry and its significance for knee health. Additionally, please discuss biomechanical sex differences, emphasizing disparities in muscle strength and alignment (e.g., valgus angle).

Thank you for this suggestion. We have addressed both points in the new "Personal and Lifestyle Determinants of Knee Health" section.

Comment 6: Page 7, lines 234-236: The authors focus primarily on flexion-extension control (walking and running). However, rotational and pivot movements are critical for dynamic stability and injury prevention. Please add a brief paragraph acknowledging that rotational and pivot motions are essential degrees of freedom influencing ligament strain and proprioceptive control, especially during cutting or change-of-direction tasks.

We appreciate this insightful comment. We agree that our initial discussion was overly focused on the sagittal plane, which does not capture the knee’s complex role in dynamic stability. To address this, we have added the following sentences to page 11, lines 303 - 312.

While the demands of walking are primarily in the sagittal plane (flexion-extension), this does not fully capture the knee's role in dynamic stability. Rotational (transverse plane) and side-to-side (frontal plane) motions must be controlled to prevent injury, especially during cutting or pivoting [110]. This dynamic stability is provided by a concert of muscular action by several accessory muscles [111]. This multi-planar neuromuscular control is important for maintaining joint congruency and preventing excessive strain on passive structures, such as the ACL [97].

Comment 7: Page 7, lines 243-245: The sensorimotor system listed the menisci as key proprioceptive structures. This is not entirely accurate. Menisci play a secondary role compared to ligaments, tendons, and muscles. Please revise this statement by removing the menisci from the list of primary proprioceptive structures.

Thank you for pointing this out. We agree and removed the menisci from the list as a primary proprioceptive structure.

Comment 8: Page 15, lines 582-596: Although the section on structural integrity covers key imaging modalities, clinical assessment parameters (pain, swelling, range of motion, joint stability, and palpation findings) should also be included. Clinical evaluation remains essential, as imaging findings alone may not correlate with symptoms (e.g., in knee OA, radiographic findings align with clinical symptoms in only about 50% of cases). Incorporating clinical and functional assessments will provide a more complete diagnostic and prognostic perspective. These aspects should also be represented in Figure 6.

Thank you for this insightful comment. We agree that clinical evaluation is crucial in understanding the symptomology and developing a treatment plan. We have updated the structural integrity domain to include a statement about the importance of clinical evaluation and incorporated it into Figure 6.

Reviewer 1 Minor Comments

  • Page 1, lines 32-34: Missing space between “time” and “[9].”

    Author Response:  We have revised to add space.
  • Page 1, lines 38-40: Too many commas; please revise the sentence for clarity.

    Author Response:  We have revised for clarity: “Compromised knee health due to acute injuries like anterior cruciate ligament (ACL) tears or chronic conditions like knee osteoarthritis (OA) disrupts the biomechanical and functional equilibrium of the knee, resulting in a cascade of detrimental effects.
  • Page 2, lines 43-44: The word “synonym” should be reviewed for appropriateness.

    Author Response: We have revised this closing without using the word 'synonymous' to state the relationship between maintaining knee health and mobility more directly.Therefore, maintaining knee health means lessening mobility restrictions and preserving mobility.”
  • Page 2, lines 45-59: These two paragraphs should be merged.

Author Response: Merged the two paragraphs.

  • Page 2, lines 73-75: “Osteoarthritis” was already abbreviated (page 1, line 39); use only “OA.”

    Author Response: Changed “osteoarthritis” to “OA”
  • Page 2, lines 80-83: “Anterior cruciate ligament” was already abbreviated (page 1, line 39); use only “ACL.”

Author Response: Changed “anterior cruciate ligament” to “ACL”

  • Page 3, lines 92-94: Use “OA” instead of the full term.

Author Response: Changed “osteoarthritis” to “OA”

  • Page 3, lines 127-130: Use “OA” instead of the full term.

Author Response: Changed “osteoarthritis” to “OA”

  • Page 3, lines 134-137: Spell out “United States” before later abbreviating as “U.S.”
    Author Response: Changed “US” to “United States”
  • Page 5, lines 191-194: Use “ACL” instead of the full term.

Author Response: Changed “anterior cruciate ligament” to “ACL”

  • Page 5, lines 196-199: Missing space between “degeneration” and “[52].”

Author Response:  We have revised to add space.

  • Page 5, lines 199-201: Missing space between “walking” and “[11].”

Author Response:  We have revised to add space.

  • Page 7, lines 231-233: Missing space between “(LCL)” and “[77].”

Author Response:  We have revised to add space.

  • Page 7, lines 243-246: Missing space between “proprioception” and “[82, 83].”

Author Response:  We have revised to add space.

  • Page 8, line 279: Missing space between “=” and “Posterior.”

Author Response:  We have revised to add space.

  • Page 10, lines 239-240: Missing space between “muscles” and “[59].”

Author Response:  We have revised to add space.

  • Page 10, lines 346-351: Missing space between “(see Figure 5A)” and “[57].”

Author Response:  We have revised to add space.

  • Page 11, lines 372-376: Missing space between “(the sum of moments from the ankle, knee, and hip)” and “[75].”

Author Response:  We have revised to add space.

  • Page 12, lines 413-416: Excessive spacing between “39%” and “in.”

Author Response:  We have revised to remove extra space.

  • Page 13, lines 445-449: Use “OA” instead of the full term.

Author Response:  We have revised to add space.

  • Page 13, lines 467-469: Missing space between “dysfunction” and “[26, 102].”

Author Response:  We have revised to add space.

  • Page 13, lines 469-472: Missing space between “weakness” and “[115].”

Author Response:  We have revised to add space.

  • Page 13, lines 487-489: Use “OA” instead of the full term; also, in the end of the sentence is “see [118-120]” – appears incomplete; please check for missing text.

Author Response:  Changed “osteoarthritis” to “OA”. We clarified this sentence to highlight the relevant reviews.

  • Page 13, lines 491-493: Missing space between “joint” and “[121, 122].”

Author Response:  We have revised to add space.

  • Page 14, lines 512-516: Missing spaces between “cytokines” and “[124],” “species” and “[129],” and “breakdown” and “[120, 130].”

Author Response:  We have revised to add spaces.

  • Page 14, lines 516-519: Missing space between “modeling” and “[86, 131].”

Author Response:  We have revised to add space.

  • Page 14, lines 522-524: Excessive spacing between “[120, 130].” and “Ligaments.”

Author Response:  We have revised to remove extra space.

  • Page 14, lines 526-528: Missing space between “patterns” and “[118-120].”

Author Response:  We have revised to add space.

  • Page 16, lines 593-596: Misplaced “=” symbol.

Author Response:  Removed “=”.

  • Page 16, lines 598-601: Missing space between “coordination” and “[15, 19, 20, 29, 30, 109, 600 144-149].”

Author Response:  We have revised to add space.

  • Page 18, lines 661-661: Excessive space between “acknowledges” and “the.”

Author Response:  We have revised to remove extra space.

Reviewer 2 Report

Comments and Suggestions for Authors

Abstract

Line 10: “Impairments in knee integrity…”

I would suggest not using integrity here as you use it to describe structural status later in the abstract which may lead some readers to think you mean structural changes a suggestion would be to say function or something similar instead.

Introduction

Lines 40-42: “Substantial evidence suggests that 40 muscle health declines [25-29], cartilage degrades [19, 30-34], and bone remodeling occurs 41 [18, 19], which in turn alters knee biomechanics [35] and increases the mechanical and 42 metabolic demands of locomotion [36].”

A figure detailing article extraction, screening, and selection would be helpful to confirm the guidelines for a literature review were met. This could be a supplemental figure.

Author Response

Thank you very much for taking the time to review our manuscript thoroughly. We appreciate your feedback and have included your suggestions in the latest revision. Please find the detailed responses below, along with the corresponding revisions/corrections highlighted in track changes in the resubmitted files.

Abstract

Comment 1: Line 10: “Impairments in knee integrity…”

I would suggest not using integrity here as you use it to describe structural status later in the abstract which may lead some readers to think you mean structural changes a suggestion would be to say function or something similar instead.

Author Response:  Thank you for pointing this out. We agree, this could be confusing to readers. We revised to “function” instead of “integrity”.

Introduction

Comment 2: Lines 40-42: “Substantial evidence suggests that 40 muscle health declines [25-29], cartilage degrades [19, 30-34], and bone remodeling occurs 41 [18, 19], which in turn alters knee biomechanics [35] and increases the mechanical and 42 metabolic demands of locomotion [36].”

Author Response: We have revised this sentence for clarity.

“Substantial evidence suggests that muscle health declines [25-29], cartilage degrades [19,30-34], and bone remodeling occurs [18,19]. Furthermore, this alters knee biomechanics [35] and increases the mechanical and metabolic demands of locomotion [36]. Therefore, maintaining knee health means lessening mobility restrictions and preserving mobility.”

Comment 3: A figure detailing article extraction, screening, and selection would be helpful to confirm the guidelines for a literature review were met. This could be a supplemental figure.

Author Response:  We appreciate this valuable suggestion and agree with the reviewer on the importance of methodological transparency. We have opted not to include a PRISMA-style flow diagram, as we believe it would misrepresent the scope and methodology of this manuscript. A PRISMA diagram is the standard for a systematic review, which aims to find and synthesize all available evidence to answer a specific quantitative question. As this is a narrative review, our primary goal was to synthesize foundational, high-impact, and contemporary literature to provide a comprehensive overview and propose a new theoretical model (the IKHMM).

However, to directly address the reviewer's excellent point about transparency, we have added a comprehensive Methods section (Page 2, Lines 57-77). This new section provides an explicit description of our systematic search strategy, the databases used, our inclusion/exclusion criteria, and the process for prioritizing high-quality studies.

Round 2

Reviewer 1 Report

Comments and Suggestions for Authors

Dear Authors,

Thank you for the revisions made so far. I appreciate the time and effort you put into addressing my previous comments. Most points have been resolved, but a few items still require attention:

  1. I could not find the modification related to my previous comment number 8. Please indicate clearly where this change was made in the manuscript.

  2. On page 2, lines 65–66, you state: “Additionally, the Research Rabbit AI tool was employed to visualize connections between papers and pinpoint key articles.” Please clarify that it was used as a bibliometric assistance tool.

  3. In the Methods section, the " " related to the selected keywords appear in different formats. Please revise to ensure consistent formatting.

  4. The subsection 4.3 Sex and Genetic Factors should be formatted in italics.

  5. In section 6 (The Knee as a Biomechanical Nexus in the Kinetic Chain), please include information regarding how limb length asymmetry can influence knee injuries or pathologies. Several studies support this relationship (for example: Murray KJ, Azari MF. Leg length discrepancy and osteoarthritis in the knee, hip and lumbar spine. J Can Chiropr Assoc. 2015;59(3):226-37.; Harvey WF, et al. Association of leg-length inequality with knee osteoarthritis. Ann Intern Med. 2010;152(5):287–295.; Golightly YM, et al. Relationship of limb length inequality with radiographic knee and hip osteoarthritis. Osteoarthr Cartilage. 2007;15(7):824–829.; Golightly YM, et al. Symptoms of the knee and hip in individuals with and without limb length inequality. Osteoarthr Cartilage. 2009;17(5):596–600.; Resende RA, et al. Mild leg length discrepancy affects biomechanics in individuals with knee osteoarthritis during gait. Clin Biomech. 2016;38:1–7.).

  6. Throughout the manuscript, the expression et al. is used inconsistently. Sometimes it appears as et al., and other times as et al.’s. Please revise for consistency.

Thank you again for your continued work on this manuscript. I look forward to reviewing the revised version.

Author Response

Response to Reviewer 1 Comments

1. Summary

Thank you once more. We value your careful attention and feedback, which we have incorporated into the latest revision. Below, you'll find detailed responses along with the highlighted revisions and corrections in the resubmitted files.

2. Questions for General Evaluation

Reviewer’s Evaluation

Response and Revisions

Does the introduction provide sufficient background and include all relevant references?

Yes/Can be improved/Must be improved/Not applicable

[Please give your response if necessary. Or you can also give your corresponding response in the point-by-point response letter. The same as below]

Are all the cited references relevant to the research?

Yes/Can be improved/Must be improved/Not applicable

Is the research design appropriate?

Yes/Can be improved/Must be improved/Not applicable

Are the methods adequately described?

Yes/Can be improved/Must be improved/Not applicable

Are the results clearly presented?

Yes/Can be improved/Must be improved/Not applicable

Are the conclusions supported by the results?

Yes/Can be improved/Must be improved/Not applicable

3. Point-by-point response to Comments and Suggestions for Authors

Comment 1: I could not find the modification related to my previous comment number 8. Please indicate clearly where this change was made in the manuscript.

Author Response: The response to the comment was unintentionally omitted in the last revision. We sincerely apologize and have added the following statement in Section 7.1.1 Domain 1: Structural Integrity, specifically at lines 651–655.

“However, imaging findings alone may not fully capture the clinical picture. Studies show that radiographic severity correlates with patient-reported symptoms in only about 50% of cases [162], underscoring the need to integrate clinical assessments, such as pain, swelling, range of motion, and joint stability, for a more comprehensive diagnostic and prognostic perspective.”

Comment 2: On page 2, lines 65–66, you state: “Additionally, the Research Rabbit AI tool was employed to visualize connections between papers and pinpoint key articles.” Please clarify that it was used as a bibliometric assistance tool.

Author Response: Thank you for bringing this to our attention. We have added the following correction to the Methods section at lines 65-66:

Author Response: Thank you for bringing this to our attention. We have added the following correction to the Methods section at lines 65-66 to clarify the use of this tool:

“Research Rabbit, a bibliometric assistance tool, was used to visualize connections between research articles and pinpoint additional key articles.” 

Comment 3: In the Methods section, the " " related to the selected keywords appear in different formats. Please revise to ensure consistent formatting.

Author Response: We have ensured consistent formatting for the keywords in the Methods section.

Comment 4: The subsection 4.3 Sex and Genetic Factors should be formatted in italics.

Author Response: We have italicized this subsection.

Comment 5: ), please include information regarding how limb length asymmetry can influence knee injuries or pathologies. Several studies support this relationship (for example: Murray KJ, Azari MF. Leg length discrepancy and osteoarthritis in the knee, hip and lumbar spine. J Can Chiropr Assoc. 2015;59(3):226-37.; Harvey WF, et al. Association of leg-length inequality with knee osteoarthritis. Ann Intern Med. 2010;152(5):287–295.; Golightly YM, et al. Relationship of limb length inequality with radiographic knee and hip osteoarthritis. Osteoarthr Cartilage. 2007;15(7):824–829.; Golightly YM, et al. Symptoms of the knee and hip in individuals with and without limb length inequality. Osteoarthr Cartilage. 2009;17(5):596–600.; Resende RA, et al. Mild leg length discrepancy affects biomechanics in individuals with knee osteoarthritis during gait. Clin Biomech. 2016;38:1–7.).

Author Response: We appreciate this valuable suggestion and agree with the reviewer on the importance of leg length asymmetry. We have added the following paragraph to the opening section of The Knee as a Biomechanical Nexus in the Kinetic Chain in lines 302 - 312:

“Importantly, not all challenges to knee function arise from within the joint itself. Structural asymmetries stemming from the femur or tibia can impose external biomechanical demands that hinder the knee’s ability to absorb ground reaction forces during gait, contributing to persistent clinical symptoms [96,97]. Leg length inequality or discrepancies greater than 20mm (2cm), whether measured anthropometrically with a tape measure or radiographically using digital rulers, is associated with increased pain, stiffness, aching, and joint degeneration at the knee [96,98-100]. Notably, current evidence suggests that the knee may be the initial site of degradation within the kinetic chain, precipitating compensatory degradation in the hip or lower back [97,100,101]. Leg length asymmetry affects both adolescents returning to sport after ACLR [99] and older adults with knee OA [98], underscoring its significance as a critical factor in the knee’s kinetic role.”

Comment 6: Throughout the manuscript, the expression et al. is used inconsistently. Sometimes it appears as et al., and other times as et al.’s. Please revise for consistency.

Author Response: We have corrected the et al.'s to show possession correctly.

Round 3

Reviewer 1 Report

Comments and Suggestions for Authors

Dear authors,

Thank you again for providing the revised version of your manuscript. You successfully changed the manuscript according to my suggestions. I have no further comments. 

Great job and best of luck!